# Fibrinogen-Induced Regeneration Sealing Technique (F.I.R.S.T.): A Retrospective Clinical Study on 105 Implants with a 3–7-Year Follow-Up

**DOI:** 10.3390/jcm13226916

**Published:** 2024-11-17

**Authors:** Márton Kivovics, Vincenzo Foti, Yaniv Mayer, Eitan Mijiritsky

**Affiliations:** 1Department of Public Dental Health, Semmelweis University, Szentkirályi utca 40, 1088 Budapest, Hungary; 2Private Practice in Periodontology and Implantology, 16129 Genova, Italy; drvincenzofoti@gmail.com; 3Department of Periodontology, School of Graduate Dentistry, Rambam Health Care Campus (RHCC), HaAliya HaShniya St. 8, Haifa 3109601, Israel; yaniv.mayer@technion.ac.il; 4Faculty of Medicine, Technion—Israel Institute of Technology, Efron St. 1, Haifa 3525433, Israel; 5Department of Head and Neck Surgery and Maxillofacial Surgery, Tel-Aviv Sourasky Medical Center, School of Medicine, Tel Aviv University, 6 Weizmann Street, Tel-Aviv 6423906, Israel; mijiritsky@bezeqint.net; 6Goldschleger School of Dental Medicine, Faculty of Medicine, Tel Aviv University, Tel Aviv 6934228, Israel

**Keywords:** bone graft, xenograft, guided bone regeneration (GBR), fibrin sealant, cortical lamina, immediate implant placement, alveolar ridge preservation (ARP), vertical bone augmentation, horizontal bone augmentation, cone beam computed tomography (CBCT)

## Abstract

**Background/Objectives:** The primary aim of this retrospective clinical study was to assess the success and bone gain achieved by using the Fibrinogen-Induced Regeneration Sealing Technique (F.I.R.S.T.) in different indications. **Methods:** In this single-center retrospective clinical study, F.I.R.S.T. was performed in the following indications: alveolar ridge preservation (ARP), immediate implant placement, and horizontal and vertical guided bone regeneration (GBR) with simultaneous dental implant placement. F.I.R.S.T. is a modified approach to GBR characterized by the application of a porcine cortical lamina, as a long-term resorbable bone barrier to cover the bone defect, and a fibrin sealant for easy adaptation of the xenogenic bone graft material and the fixation of the collagenic bone barrier. Patients with uncontrolled systemic diseases, medications, or diseases that may alter bone metabolism; local inflammation; poor oral hygiene; and heavy smoking were excluded from this study. Horizontal and vertical bone gain (HBG and VBG) were measured by comparing postoperative and preoperative cone beam computed tomography (CBCT) reconstructions. Patients were recalled for controls and oral hygiene treatment every 6 months. **Results:** Altogether, 62 patients (27 male, 35 female, age 63.73 ± 12.95 years) were included in this study, and 105 implants were placed. Six implants failed during the 50.67 ± 22.18-month-long follow-up. Cumulative implant survival throughout the groups was 94.29 %. In the immediate implant group, HBG was 0.86 mm (range: −0.75–8.19 mm) at the 2 mm subcrestal level, while VBG was 0.87 ± 1.21 mm. In the ARP group, HBG was 0.51 mm (range: −0.29–3.90 mm) at the 2 mm subcrestal level, while VBG was −0.16 mm (range: −0.52–0.92 mm). In the horizontal GBR group, HBG was 2.91 mm (range: 1.24–8.10 mm) at the 2 mm subcrestal level. In the vertical GBR group, VBG was 4.15 mm (range: 3.00–10.41 mm). **Conclusions:** F.I.R.S.T. can be utilized successfully for bone augmentation. The vertical and horizontal bone gains achieved through F.I.R.S.T. allow for implant placement with adequate bone width on both the vestibular and oral aspects of the implant.

## 1. Introduction

For the long-term stability of soft and hard tissues surrounding dental implants and for ideal esthetics, a thickness of 1.5 mm of the bone on the vestibular and oral aspects of the implant is required [1,2], and implants should be placed 2–3 mm below the cementoenamel junction of the adjacent teeth [2,3]. However, due to alveolar atrophy, following tooth removal, the available bone may be insufficient for dental implant placement [4,5,6]. In cases of severe atrophy, a staged approach is necessary. First, bone augmentation is performed to restore the bone volume so that it is sufficient for implant placement, and after an adequate healing period, dental implants are placed in the augmented bone [7,8]. In cases where dental implants may be placed with adequate stability, bone augmentation may be performed simultaneously [8,9].

Guided bone regeneration (GBR) has become a standard in the regeneration of fenestration- and dehiscence-type defects simultaneously with dental implant placement [10]. The essence of GBR is using cell-occlusive membranes that prevent the in-growth of epithelial cells and connective tissue in the defect, allowing bone regeneration [11,12,13,14]. As part of the GBR technique, titanium-reinforced non-resorbable membranes, stock or customized titanium meshes, or particulated bone graft materials may be used for space maintenance [10]. Bone substitute materials may promote bone healing due to their osteoconductive and, in certain cases, osteoinductive properties [15,16,17,18,19]. The application of a rigid collagenic bone barrier may contribute to space maintenance [20,21,22,23,24,25].

Micro-movements during the healing process have been reported to favor connective tissue formation over the integration of bone graft materials and dental implants [26]. Titanium and resorbable pins are utilized to fix membranes and prevent movements due to external forces and muscle traction [27,28]. A disadvantage of such a method of membrane fixation may be the increased duration of surgery. Titanium pins may become loose and migrate in the surgical site, which warrants another intervention to address this complication. Fibrin sealants may be applied for particulate graft stabilization during GBR [29]. Combined with bone graft materials, a fibrin sealant provides a better scaffold that promotes bone tissue formation due to its osteoconductive and osteoinductive properties [30,31,32,33,34,35,36].

The Fibrinogen-Induced Regeneration Sealing Technique (F.I.R.S.T.) is a modified approach to simultaneous horizontal or vertical GBR [37,38,39,40]. Following the elevation of a full-thickness flap, the defect is filled with a collagenic porcine graft (OsteoBiol^®^ Gen-Os^®^, Tecnoss^®^, Giaveno, Italy). A porcine cortical lamina (OsteoBiol^®^ Soft Cor-tical Lamina^®^, Tecnoss^®^, Giaveno, Italy) is used as a long-term resorbable bone barrier to cover the bone defect. A fibrin sealant (Tisseel^®^, Baxter^®^, Deerfield, IL, USA) is added for easy adaptation of the xenogenic bone graft material and the fixation of the collagenic bone barrier. Finally, the flap is mobilized to allow tension-free, primary wound closure.

Alveolar atrophy may be prevented or significantly decreased using alveolar ridge preservation (ARP) following atraumatic tooth extraction [19,41,42,43]. Several surgical methods and biomaterials have been utilized successfully for socket grafting [11,17,18,42,44,45,46,47]. Adapting the principles of F.I.R.S.T., a modified ARP may be per-formed in case of a defect on the oral or vestibular bone wall following extraction. During the procedure, the tooth is removed atraumatically, and a full-thickness envelope flap is raised. The sockets are packed with collagenic porcine graft (OsteoBiol^®^ Gen-Os^®^, Tecnoss^®^, Giaveno, Italy). A porcine cortical lamina (OsteoBiol^®^ Soft Cor-tical Lamina^®^, Tecnoss^®^, Giaveno, Italy) is used to cover the buccal or oral bone wall defect and is fixed using fibrin sealant (Tisseel^®^, Baxter^®^, Deerfield, IL, USA). The margins of the flap are stabilized by fibrin sealant without flap mobilization, leaving the wound intentionally open for secondary healing, as the antibacterial effect of the fibrin sealant allows the soft tissues to close [16]. Defective vestibular and oral bone walls may be treated similarly during immediate implant placement using the xenograft to fill the gap between the restored bone wall and the dental implant.

The primary aim of this retrospective clinical study was to assess the success and bone gain achieved by using F.I.R.S.T. in horizontal and vertical bone augmentation, performed simultaneously with dental implant placement, immediate implant placement, and ARP. The secondary purpose of this study was to evaluate the survival and the rate of biological complications of the implants placed using this modified GBR approach in the mid-term.

## 2. Materials and Methods

### 2.1. Study Design

This single-center retrospective, observational, clinical study was conducted in accordance with the Helsinki Declaration. The surgical procedures and the biomaterials used in this study were thoroughly explained to the patients enrolled. Patients signed informed consent documents regarding the surgical interventions and the use of their healthcare-related data. Patients from a private practice were included in this study. The reporting of this study conforms to the STROBE statement. The STROBE checklist is presented in Appendix A.

Inclusion criteria:Patients over the age of 18 years.Patients who received implant-borne prostheses for oral rehabilitation.Patients who required bone augmentation at the time of implant placement.

Exclusion criteria:Major systemic disease that contraindicates minor surgery.Psychiatric contraindications.Patients on medication interfering with bone metabolism (steroid therapy, antiresorptive medication, etc.).Radiation to the head or neck region within the previous five years.Localized periapical disease and cysts.Evidence of uncontrolled periodontal disease.Alcohol Use Disorder or recreational drug abuse.Heavy smoking (>10 cigarettes/day).Pregnancy or nursing.Poor oral hygiene.Unavailability for regular follow-ups.

### 2.2. Surgical Interventions

#### 2.2.1. Horizontal and Vertical GBR Using F.I.R.S.T.

Horizontal GBR using F.I.R.S.T. was opted for when dental implants could be placed with sufficient stability in the recipient sites and the thickness of the bone on the vestibular and oral aspects of the implant was less than 1.5 mm. Vertical GBR using F.I.R.S.T. was performed when, upon placing the implants in the recipient sites at the prosthetically correct depth, sufficient stability could be achieved, and vertical augmentation was necessary so that the implants would be surrounded by adequate bone.

Before the intervention, the collagenic cortical lamina (OsteoBiol^®^ Soft Cortical Lamina^®^, Tecnoss^®^, Giaveno, Italy) was placed in sterile saline solution for hydration for 5 min, and the fibrin sealant was prepared according to the manufacturer’s instructions. The thrombin component of the fibrin sealant was diluted in a ratio of 1:10 with sterile double-distilled water. Patients rinsed with a 0.2% chlorhexidine solution for 1 min. All interventions were performed by an experienced surgeon (V.F.) under local anesthesia using 4% Articaine with adrenaline in a ratio of 1:200,000. A mucoperiosteal flap was raised from a crestal incision with mesial and distal relieving incisions. The bone was decorticalized using a Lindemann bur to allow revascularization of the augmented sites. Dental implants (TSIII SA, Osstem, Seoul, South Korea) were placed in the prosthetically correct position. A porcine collagenic xenograft (OsteoBiol^®^ Gen-Os^®^, Tecnoss^®^, Giaveno, Italy) was applied in combination with the cortical lamina (OsteoBiol^®^ Soft Cortical Lamina^®^, Tecnoss^®^, Giaveno, Italy) to perform GBR. The fibrin sealant (Tisseel^®^, Baxter^®^, Deerfield, IL, USA) was added to obtain a putty consistency when working with the xenogenic bone graft material and to fix the cortical lamina on the surface of the bone. The full-thickness flap was mobilized to allow tension-free primary closure. The flap margins were stabilized with non-resorbable 5/0 Polypropylene sutures (Aragò, Barcelona, Spain). Patients were prescribed Amoxicillin 500 mg every 8 h for 7 days and Ibuprofen 600 mg 3 times a day for 3 days. Patients were instructed to use Chlorhexidine 0.2% rinses in the morning and the evening for 7 days. The sutures were removed after 2 weeks. Implant uncovering was performed after 6 months of healing under local anesthesia.

In cases when the “One-time Cortical Lamina” technique was used, the flaps were not mobilized; flap margins were sutured around the healing abutments and a second intervention for implant uncovering could be avoided. F.I.R.S.T. for horizontal and vertical augmentation is presented in Figure 1 and Figure 2.

#### 2.2.2. Immediate Implant Placement Using F.I.R.S.T.

In cases where the available bone beyond the apex of the tooth was sufficient, primary stability could be achieved, and the implant could be placed in the prosthetically correct position, atraumatic extraction was performed, and immediate implant placement was carried out. In the absence of either of these criteria, ARP and delayed implant placement were performed.

The surgical site was accessed using the tunnel technique. A collagenic porcine cortical lamina (OsteoBiol^®^ Soft Cortical Lamina^®^,Tecnoss^®^) was adapted to bone dehiscence or fenestration on the vestibular or oral socket wall. The gap between the cortical lamina and the implant was grafted using collagenic porcine bone (OsteoBiol^®^ Gen-Os^®^, Tecnoss^®^). After 1:10 dilution of the thrombin component with sterile double-distilled water, the fibrin sealant (Tisseel^®^, Baxter^®^, Deerfield, IL, USA) was applied to fix the bone graft material in the gap and the cortical lamina on the surface of the bone. The margins of the flap were stabilized by fibrin sealant around the healing abutment of the implant or the provisional crown. The immediate implant placement performed with F.I.R.S.T. is presented in Figure 3.

#### 2.2.3. ARP and Delayed Implant Placement Using F.I.R.S.T.

Following the preparation of the biomaterials, periotomes and luxators were used for the detachment of the periodontal ligaments, and atraumatic tooth removal was performed with forceps. After thorough debridement of the socket, a full-thickness envelope flap was raised. A collagenic porcine cortical lamina (OsteoBiol^®^ Soft Cortical Lamina^®^, Tecnoss^®^) was trimmed to extend beyond the borders of the dehiscence or fenestration defect on the oral or vestibular bone wall by at least 3 mm and was placed beneath the envelope flap. A porcine collagenic xenograft (OsteoBiol^®^ Gen-Os^®^, Tecnoss^®^, Giaveno, Italy) was packed into the alveolus using light force. After 1:10 dilution of the thrombin component with sterile double-distilled water, the fibrin sealant (Tisseel^®^, Baxter^®^, Deerfield, IL, USA) was applied to fix the bone graft material in the socket and the cortical lamina on the surface of the bone. No sutures were applied, and the bone graft material was left intentionally exposed. F.I.R.S.T. for ARP is presented in Figure 4.

The postoperative medications and instructions were similar to those following the GBR procedures. After a 6-month-long healing time, under local anesthesia, dental implants were placed in the grafted sockets non-submerged.

### 2.3. Outcome Measures

For this study, the primary outcome measures were as follows:The success of the surgical interventions. The bone augmentation was deemed a successful one if a healthy soft and hard tissue situation could be achieved and implants were surrounded by at least 1.5 mm bone on the vestibular and oral sides.Horizontal bone gain (HBG) and vertical bone gain (VBG)

Preoperative cone beam computed tomography (CBCT) scanning was performed to assess the anatomy, dimensions, and possible pathologies of the alveolar ridges using Planmeca Viso G7 hardware (Planmeca, Helsinki, Finland). The scanning conditions were a 140 mm × 100 mm field of view, a 150 µm voxel size, a 100 kV tube voltage, and a 63 mAs charge. After a healing time of 8 months, a postoperative CBCT scan was performed with the same conditions. Identical landmarks were used to trace identical panoramic curves on post- and preoperative CBCT reconstructions. Orthoradial sections in the middle of the implants were chosen on the postoperative CBCT, and their mesiodistal distance from the distalmost point at the bone level of the last tooth in the quadrant was applied to identify the same section on the preoperative CBCT reconstruction. An anatomical landmark (e.g., base of the mandible, horizontal level of the palate, base of the maxillary sinus…) was chosen to measure pre- and postoperative bone height, and the preoperative bone height was subtracted from the postoperative height to calculate the VBG in millimeters. Pre- and postoperative bone width was measured 2 mm and 5 mm below the highest level of the alveolar ridge. To calculate the HBG in millimeters, the preoperative width at both the 2 mm and 5 mm subcrestal levels was subtracted from the postoperative bone width.

The secondary outcome measures addressed bone gain and the health of the soft and hard tissues surrounding the dental implants placed in the augmented bone.

Implant survival:

The implant is present at the recipient site at the time of follow-up.

Prevalence of peri-implant mucositis:

Peri-implant mucositis was diagnosed if signs of inflammation (redness, swelling, line or drop of bleeding within 30 s following probing) could be observed on the peri-implant soft tissues with no additional bone loss following initial healing [48,49].

Prevalence of peri-implantitis:

Peri-implantitis was diagnosed if the radiographically determined vertical bone loss was greater than 3 mm in combination with bleeding on probing (BOP) and probing depths (PD) greater than 6 mm [49,50].

### 2.4. Statistical Analysis

The quantitative data were expressed as standard descriptive statistics. A Shapiro–Wilk-test was performed to assess whether the data were normally distributed. For outcome measures that follow a normal distribution, the means and standard deviations were reported. In cases of non-normal distribution of the data, the median and range were displayed. Descriptive statistical analyses were conducted using SPSS 25.0 software (IBM Corporation, New York, NY, USA).

## 3. Results

Altogether, 62 patients (27 male, 35 female, age 63.73 ± 12.95 years) were included in this retrospective clinical study and 105 dental implants were placed. F.I.R.S.T. was performed for immediate implant placement on six patients (three male, three female; median age: 70.5 years; range: 60–88 years), ARP on fourteen patients (nine male, five female, 63.93 ± 14.06 years), horizontal GBR on thirty patients (eight male, twenty-two female, age: 63.13 ± 12.53 years), and vertical GBR on sixteen patients (nine male, seven female, age: 61.44 ± 13.81 years). The patient characteristics are presented in Appendix A. A flow diagram for this retrospective study is presented in Figure 5.

In the immediate implant and ARP groups, all bone augmentations were deemed successful. However, complications occurred in the horizontal and vertical GBR groups.

In one of the cases in the horizontal GBR group, a purulent infection occurred two months after placing an implant at site 33 (Fédération Dentaire Internationale FDI, tooth numbering). A bone sequester was removed without flap elevation to treat the infection, and systemic antibiotics were prescribed to the patient. Following re-entry, the infection was resolved, and the implant survived with 2.76 mm and 2.19 mm horizontal bone gain at subcrestal levels of 2 mm and 5 mm, respectively. In another case of horizontal GBR, following implant placement at sites 24 and 26, a circumferential bone defect with a depth of 3 mm occurred at implant site 26. A GBR surgery was performed to regenerate the bone successfully. The most severe complication in the horizontal GBR group was a case where four implants, two tilted distal and two straight mesial, were placed to rehabilitate an edentulous maxilla. A month after surgery, all implants were removed due to mobility. However, as no infection was evident, after three months, four implants were placed in the sufficiently augmented bone. In the horizontal GBR group, 27 out of 30 interventions were deemed successful. However, in the remaining three cases, complications were managed by sequestrotomy, secondary bone augmentation, and secondary implant placement, respectively. Satisfactory results could be achieved with dental implant-borne prostheses in all cases.

In one of the cases in the vertical GBR group, two implants were placed in the atrophied alveolar ridge at sites 44 and 46. After three days, soft tissue dehiscence was observed, and due to the resulting infection, the bone graft and the implants were lost. After three months of healing, vertical GBR with F.I.R.S.T. was performed, and implants were placed successfully. In another case, vertical GBR was performed simultaneously with implant placement at site 16. A small dehiscence on the flap was observed at the time of suture removal, which healed spontaneously after one month without graft and implant loss. In the vertical GBR group, 15 out of 16 interventions were deemed successful. However, satisfactory results could be achieved following repeated intervention using the same approach.

Altogether, 105 implants were placed in this study (six in the immediate implant, 14 in the ARP, 58 in the horizontal GBR, and 27 in the vertical GBR group) and six implants failed during the 50.67 ± 22.18-month-long follow-up. All cases of implant failure could be classified as early implant failure. Cumulative implant survival was 94.29% throughout the groups, 100% in the immediate implant and ARP groups, 93.10% in the horizontal GBR group, and 92.59% in the vertical GBR group. The raw data for this study are presented in Appendix A.

In the immediate implant group, HBG was 0.86 mm (range: −0.75–8.19 mm) and 0.99 ± 0.77 mm at the 2 mm and 5 mm subcrestal levels, respectively, while VBG was 0.87 ± 1.21 mm.

In the ARP group, HBG was 0.51 mm (range: −0.29–3.90 mm) and 1.18 mm (range: 0.46–6.30 mm) at the 2 mm and 5 mm subcrestal levels, respectively, while VBG was −0.16 mm (range: −0.52–0.92 mm).

In the horizontal GBR group, HBG was 2.91 mm (range: 1.24–8.10 mm) and 3.24 mm (range: 0.53–10.65 mm) at the 2 mm and 5 mm subcrestal levels, respectively, while VBG was −0.05 mm (range: −1.09–1.31 mm).

In the vertical GBR group, HBG was 7.43 mm (range: 1.81–12.95 mm) and 2.22 mm (range: −0.35–12.50 mm) at the 2 mm and 5 mm subcrestal levels, respectively, while VBG was 4.15 mm (range: 3.00–10.41 mm).

During the follow-up period, 12 implants (12.12%) presented with peri-mucositis, and none presented with peri-implantitis.

## 4. Discussion

García-Gonzáles et al. conducted a systematic review and meta-analysis on the bone changes that may be expected following the ARP of sockets with a compromised buccal bone wall. A horizontal bone loss of 2.37 mm (95% CI 3.07–1.67) and a vertical bone loss of 1.10 mm (95% CI = 1.36–0.85) may be expected despite ARP [51]. In our study, HBG was 0.51 mm (range: −0.29–3.90 mm) and 1.18 mm (range: 0.46–6.30 mm) at the 2 mm and 5 mm subcrestal levels, respectively. HBG in the ARP group of this study may be attributed to the cortical lamina fixed using the fibrin sealant to the compromised bone wall. The cortical lamina acts not only as a resorbable bone barrier inhibiting the ingrowth of connective tissue, but also as an osteoconductive material, which enables HBG as a result of F.I.R.S.T. Vertical bone loss was 0.16 mm (range: −0.92–0.52 mm) in this study for the ARP group, comparable to that observed by García-Gonzáles et al. [51].

Clementini et al. reviewed bone dimensional changes following immediate implant placement with simultaneous regenerative procedures [52]. According to their results, a horizontal bone loss of 1.02 mm (95% CI = −0.92 to −1.12) was observed following immediate implants with a bone graft, and a vertical bone gain of 1.09 mm (95% CI = −4.04 to 6.21) may be expected following the use of a resorbable membrane and bone graft for immediate implant placement. In this study, due to the use of the collagenic cortical lamina, HBG was 0.86 mm (range: −0.75–8.19 mm) and 0.99 ± 0.77 mm at the 2 mm and 5 mm subcrestal levels, respectively. A VBG of 0.87 ± 1.21 mm was observed in this study, which is comparable to the outcomes reported by Clementini et al. [52].

According to the systematic review and meta-analysis of Elnayef et al., 3.61 ± 0.27 mm HBG may be expected as a result of simultaneous horizontal augmentation performed using GBR procedures, which is comparable to the HBG observed in this study (2.91 mm (range: 1.24–8.10 mm) and 3.24 mm (range: 0.53–10.65 mm) at the 2 mm and 5 mm subcrestal levels, respectively) [53].

According to the systematic review and meta-analysis of Urban et al., the VBG achieved by vertical augmentation using GBR with simultaneous implant placement was 3.81 mm (95% CI = 3.31–4.30) [54]. The increased VBG observed in the vertical GBR group in this study (4.15 mm (range: 3.00–10.41 mm)) may be the result of F.I.R.S.T., which enables the clinician to perform better fixation of the collagenic cortical lamina and the xenogenic bone graft material in the early stages of healing, thus avoiding micromovements that compromise bone healing in the vertically maintained space.

GBR is characterized by using membranes to inhibit the ingrowth of connective tissue end epithelium in the space designated for bone regeneration [12,13,14]. Collagen membranes in combination with particulated bone graft materials have a limited space maintenance capacity, as pressure on the tissues may compress the augmented area. Therefore, such a combination of biomaterials is primarily applied for self-containing defects. Titanium-reinforced polytetrafluoroethylene (PTFE) membranes and titanium meshes combined with collagen membranes have been suggested to increase the barrier’s capacity to maintain space, so that more extensive vertical and horizontal augmentations may be performed [10]. However, if opted for, titanium meshes and non-resorbable membranes have to be removed with a further surgical intervention following bone regeneration of the augmented site [10,14]. During F.I.R.S.T., a rigid bone barrier with sufficient space-maintaining capacity is applied as a membrane [10,11,12,13,14,15,16,17,18,19,20,21,22,23,24,25]. The barrier and the particulated bone graft material are fixed to the bone using a fibrin sealant, guaranteeing the immobility of the site during healing [37,38,39,40]. No further surgical intervention is required, as all biomaterials applied as part of F.I.R.S.T. are resorbed or integrated during healing [37,38,39,40]. According to the results of this study, F.I.R.S.T. can be used successfully for the GBR of non-self-containing defects, such as ARP and immediate implant placement in case the bone walls are compromised, as well as vertical and extensive horizontal augmentation. The Increased HBG observed in the ARP and immediate implant placement groups compared to the values reported in the literature may be attributed to the osteoconductive properties of the xenogenic cortical lamina used during surgery [47,48]. The collagenic cortical lamina promotes healing not only by preventing connective tissue and epithelium migration into the space maintained for bone healing, but also by remodeling into cortical bone. Increased the VBG observed in the vertical GBR group in this study compared to the results reported in the literature is enabled by the fixating effect of the fibrin sealant. Fixing the cortical lamina and the particulated porcine collagenic xenograft inhibits micromovements that promote bone healing.

Throughout the literature, various definitions have been applied for peri-implantitis. Several studies have determined that 0.5 mm is the threshold of peri-implant bone loss above which peri-implantitis is diagnosed. Consequently, meta-analyses present wide ranges for peri-implantitis and peri-implant mucositis. According to Salvi et al., the prevalence of peri-implant mucositis is 43% (range: 19–65%) and the prevalence of peri-implantitis is 22% (range: 1–47%) [55,56]. However, several cohort studies report a lower prevalence of biological complications due to the strict exclusion of compromised patients and a thorough maintenance program [57,58,59]. In our study, a low prevalence of peri-implant mucositis and no cases of peri-implantitis were observed, which may be the result of applying the diagnostic criteria based on the Consensus report of workgroup 4 of the 2017 world workshop on the classification of periodontal and peri-implant diseases and conditions [49].

The limitations of this retrospective clinical study were the low number of cases in the immediate implant and ARP groups and the limited follow-up. Another limitation of this study is that the linear measurements of HBG and VBG were performed manually. Volumetric bone changes could have been an informative outcome measure to assess. The automatic assessment of bone gain may have decreased human error. A potential area for further research is conducting multi-center randomized clinical trials that compare F.I.R.S.T. to traditional GBR to assess its effectiveness. This is the first study to present the results of mid-term follow-up regarding F.I.R.S.T. used in multiple indications.

## 5. Conclusions

Within the limitations of this study, it can be concluded that F.I.R.S.T. can be utilized successfully for vertical and horizontal GBR with simultaneous implant placement. F.I.R.S.T. can be applied successfully for immediate implant placement and ARP in cases where a vestibular or oral bone defect is observed following atraumatic tooth removal. Vertical and horizontal bone gains achieved through F.I.R.S.T. in these indications allow for implant placement with an adequate bone width (at least 1.5 mm) on both the vestibular and oral aspects of the implant. Sufficient quality and quantity of hard and soft tissues surrounding the implants, combined with regular and thorough maintenance, prevent biological complications in the mid-term.

## Figures and Tables

**Figure 1 jcm-13-06916-f001:**
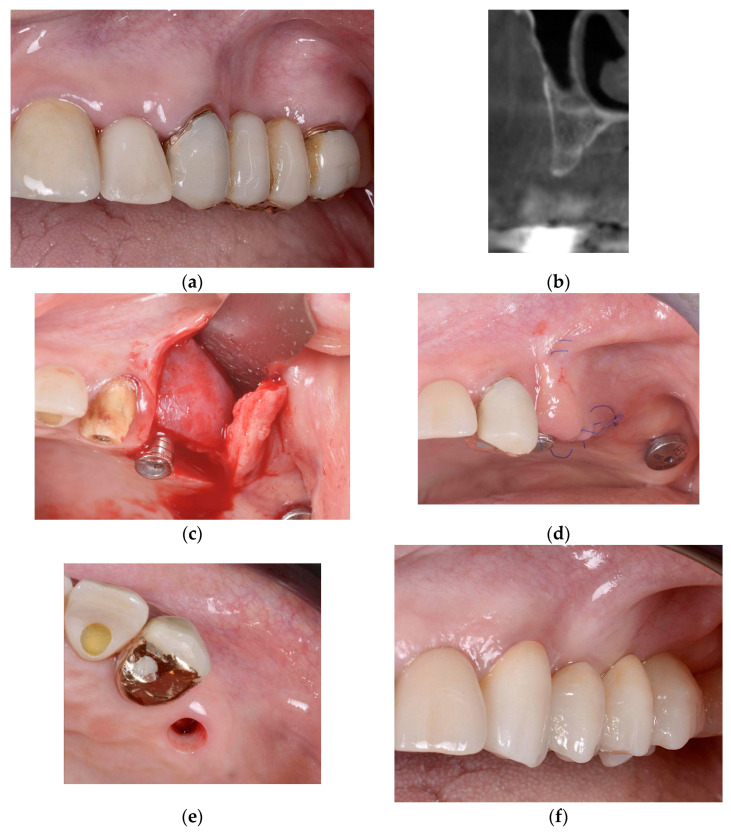
Horizontal GBR performed using F.I.R.S.T. at site 24 (Fédération Dentaire Internationale FDI, tooth numbering). (**a**) Preoperative clinical situation; (**b**) orthoradial section of the preoperative CBCT; (**c**) adaptation of the porcine collagenic xenograft and collagenic cortical lamina fixed by the fibrin sealant for horizontal augmentation; (**d**) wound closure around the healing abutment; (**e**) removed healing abutment after 6 months; (**f**) prosthesis delivery; (**g**) orthoradial section of postoperative CBCT; (**h**) control intraoral X-ray after 59 months.

**Figure 2 jcm-13-06916-f002:**
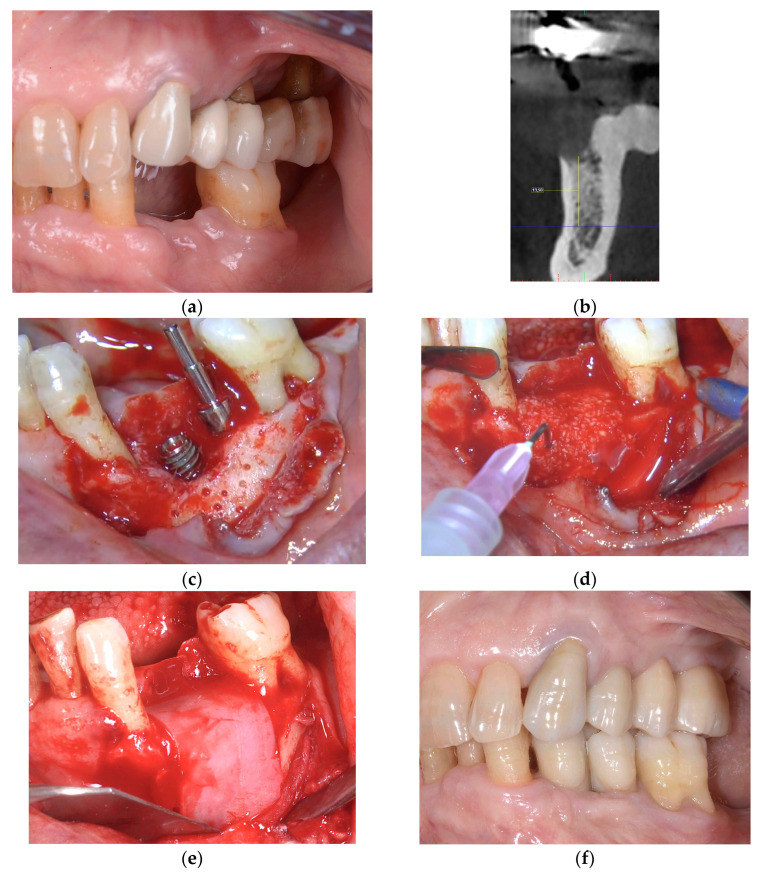
Vertical GBR performed using F.I.R.S.T. at sites 34 and 35 (Fédération Dentaire Internationale FDI, tooth numbering). (**a**) Preoperative clinical situation; (**b**) orthoradial section of the preoperative CBCT; (**c**) decorticalization of the recipient bone and dental implant placement; (**d**) application of the fibrin sealant to the porcine collagenic xenograft; (**e**) collagenic cortical lamina fixed by the fibrin sealant; (**f**) prosthesis delivery; (**g**) orthoradial section of the postoperative CBCT; (**h**) control intraoral X-ray after 60 months.

**Figure 3 jcm-13-06916-f003:**
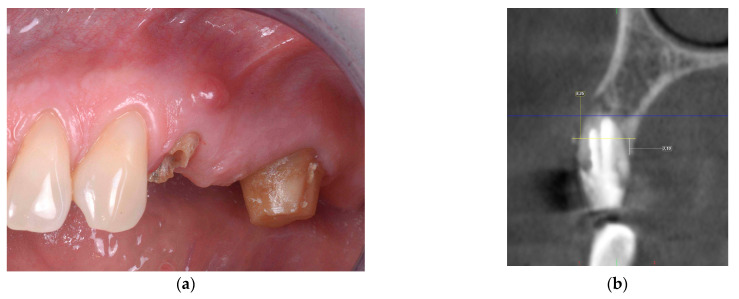
Immediate implant placement performed using F.I.R.S.T. at site 24 (Fédération Dentaire Internationale FDI, tooth numbering). (**a**) Preoperative clinical situation; (**b**) orthoradial section of the preoperative CBCT; (**c**) adaptation of the collagenic cortical lamina on the buccal bone defect; (**d**) filling the gap with porcine collagenic xenograft and application of the fibrin sealant; (**e**) prosthesis delivery; (**f**) orthoradial section of the postoperative CBCT; (**g**) control intraoral X-ray after 24 months.

**Figure 4 jcm-13-06916-f004:**
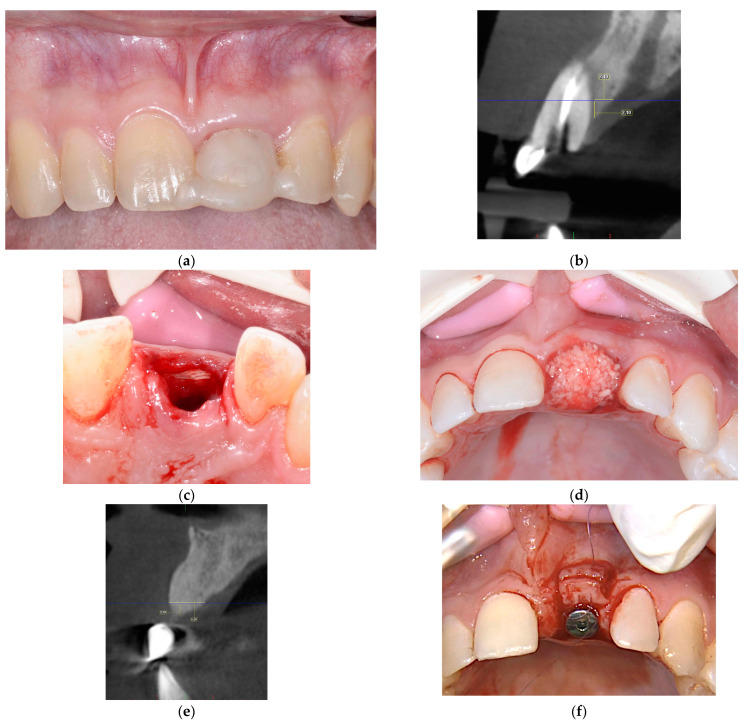
Delayed implant placement following ARP performed using F.I.R.S.T. at site 21 (Fédération Dentaire Internationale FDI, tooth numbering). (**a**) The preoperative clinical situation; (**b**) orthoradial section of the preoperative CBCT; (**c**) adaptation of the collagenic cortical lamina on the labial bone defect; (**d**) packing the porcine collagenic xenograft into the extraction socket and application of the fibrin sealant; (**e**) an orthoradial section of the postoperative CBCT 8 months post-op; (**f**) dental implant placement and Connective Tissue Graft; (**g**) prosthesis delivery; (**h**) control intraoral X-ray after 48 months.

**Figure 5 jcm-13-06916-f005:**
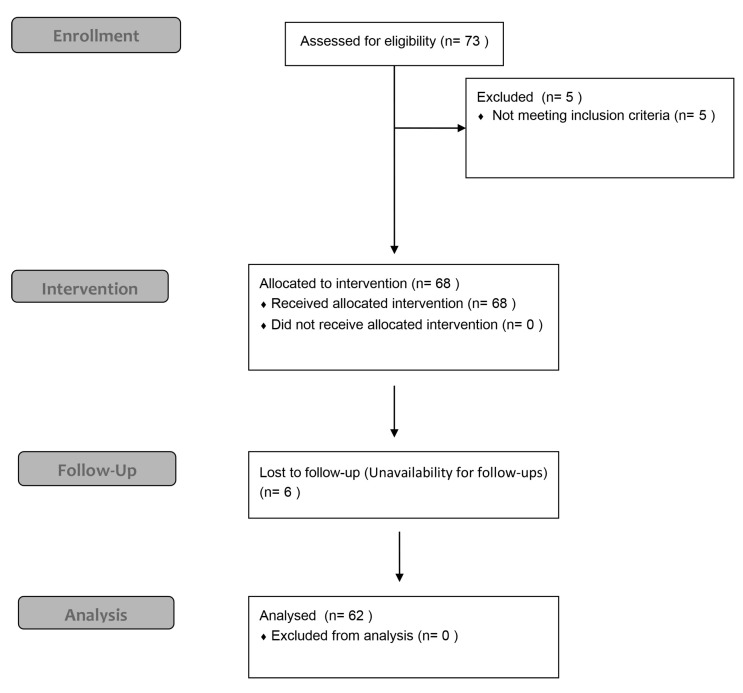
A flowchart presenting the number of individuals at each stage of this retrospective study.

## Data Availability

The data are contained within the Appendix A.

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
