# Peer review of "Fibrinogen-Induced Regeneration Sealing Technique (F.I.R.S.T.): A Retrospective Clinical Study on 105 Implants with a 3–7-Year Follow-Up"

_jcm, 2024, doi:10.3390/jcm13226916_

Round 1
Reviewer 1 Report
Comments and Suggestions for Authors
About the abstract, this article presents a bone regenerative technique based on the FIRST technique. The methodology is inadequate since it is a single-center and retrospective study, which does not allow ideal controlling for all confounding variables. The cases' typology is varied, although they have bone regeneration in common. The abstract does not indicate the exclusion criteria, technique, or follow-up times. Nor is anything shown about the statistical aspects. The result refers to the number of implants placed but should refer to the number of surgeries performed. The data on failed implants is irrelevant since the study's objective is to evaluate bone regeneration. The results in the abstract are somewhat odd since, although the gains in the surgery groups are indicated, these groups are not identified in material and method (in the abstract). In addition, significant differences between the objective data for each group are not indicated. The conclusions are inadequate since they only show a bone increase in the four groups without indicating the overall data. Therefore, the summary must be completely restructured.
In the article, the material and method identified within the surgical interventions are vertical and horizontal bone regeneration using FIRST, the placement of immediate implants using FIRST, and the preservation of the alveolus using FIRST. However, in the measurement variables, the main objective is the success of the surgical intervention. This is an inappropriate objective because it is very heterogeneous. Similarly, implant survival as a primary endpoint is inadequate for this design since not all groups have implant placement. Therefore, secondary measures, i.e., actual bone gain, should be the main objectives.
On the other hand, the different patients' characteristics in each group are not identified in tables, nor are the data identified in tables, which is very interesting to observe. Similarly, significant differences between the different groups are not identified. This aspect is essential to contribute.
Similarly, aspects related to ethics or aspects related to the statistical part are not indicated in the article's text.
Finally, the discussion could be greatly expanded, given the large amount of literature on guided bear regeneration. It would be interesting to discuss how and in what way each of the elements that make up the FIRST technique contributes to regeneration, that is, to sustained results.
Author Response
Reviewer 1
Corresponding author (CA): We would like to thank the reviewer for their time and effort in reviewing our manuscript and pointing out the issues that needed corrections.
Reviewer 1 (R1): About the abstract, this article presents a bone regenerative technique based on the FIRST technique. The methodology is inadequate since it is a single-center and retrospective study, which does not allow ideal controlling for all confounding variables.
(CA): Case reports and case series have been published on the surgical methodology described in this study. However, mid-term or long-term follow-up of these cases have not yet been presented. Therefore, in this manuscript, we aimed to present mid-term findings with the F.I.R.S.T. in different indications validating its reliability and feasibility. We agree with the reviewer, that a randomized controlled clinical trial comparing F.I.R.S.T. with conventional GBR as a control group in a single indication and with long-term follow-up may enable our research group to present stronger evidence in favor of this technique. This is going to be our further avenue of research. However, such a clinical trial can only be performed once the feasibility of F.I.R.S.T. is proven.
We have added the following to the limitation paragraph of the Discussion section:
“A potential area for further research is conducting multi-center randomized clinical trials that compare FIRST to traditional GBR to assess its effectiveness.”
R1: The cases' typology is varied, although they have bone regeneration in common. The abstract does not indicate the exclusion criteria, technique, or follow-up times.
The following was added to the Abstract:
“Patients with uncontrolled systemic diseases, medications or diseases that may alter bone metabolism, local inflammations, poor oral hygiene, and heavy smoking were excluded from this study.“
“F.I.R.S.T. is a modified approach to GBR characterized by the application of a porcine cortical lamina as a long-term resorbable bone barrier to cover the bone defect and a fibrin sealant for easy adaptation of the xenogenic bone graft material and the fixation of the collagenic bone barrier.”
“Patients were recalled for controls and oral hygiene treatment every 6 months.”
“…during the 50.67 ± 22.18 month-long follow-up”
R1: Nor is anything shown about the statistical aspects.
CA: We have added a subsection to the Materials and Methods section to explain descriptive statistics performed for this study:
“2.4. Statistical analysis
The quantitative data were expressed as standard descriptive statistics. The Shapiro-Wilk-test was performed to assess whether data were normally distributed. For outcome measures that follow a normal distribution, the means and standard deviations were reported. In case of non-normal distribution of the data, the median and range were displayed. Descriptive statistical analyses were conducted using SPSS 25.0 software (IBM Corporation, New York, United States). Descriptive statistics were corrected throughout the manuscript.”
We have revised the results accordingly.
R1: The result refers to the number of implants placed but should refer to the number of surgeries performed. The data on failed implants is irrelevant since the study's objective is to evaluate bone regeneration.
CA: We would like to thank the reviewer for pointing out this issue. The following was added to the Results section to report the success rate of the bone augmentation procedures.:
“In the horizontal GBR group, 27 out of 30 interventions were deemed successful. However, in the remaining three cases, complications were managed by sequestrotomy, secondary bone augmentation, and secondary implant placement respectively. Satisfactory results could be achieved with dental implant-borne prostheses in all cases.”
“In the vertical GBR group, 15 out of 16 interventions were deemed successful. However, satisfactory results could be achieved with a repeated intervention using the same approach.”
R1 The results in the abstract are somewhat odd since, although the gains in the surgery groups are indicated, these groups are not identified in material and method (in the abstract).
CA We have added the following to the Materials and Methods section of the Abstract to present the different groups anaysed in this study:
“In this single-center retrospective clinical study, F.I.R.S.T. was performed in the following indications: alveolar ridge preservation (ARP), immediate implant placement, horizontal and vertical guided bone regeneration (GBR) with simultaneous dental implant placement.”
R1: In addition, significant differences between the objective data for each group are not indicated. The conclusions are inadequate since they only show a bone increase in the four groups without indicating the overall data. Therefore, the summary must be completely restructured.
CA: The present study did not aim to compare bone gains achieved in different GBR indications. Therefore, vertical and horizontal bone gain was not compared by statistical means. The type of bone augmentation is dependent on the anatomy of the edentulous ridge. Such a comparison would be confusing for the reader and would be like comparing apples to pears.
R1: In the article, the material and method identified within the surgical interventions are vertical and horizontal bone regeneration using FIRST, the placement of immediate implants using FIRST, and the preservation of the alveolus using FIRST. However, in the measurement variables, the main objective is the success of the surgical intervention. This is an inappropriate objective because it is very heterogeneous.
CA: We agree that the success of the surgical intervention is heterologous as the indications of bone augmentation and that the groups are heterologous. However, success of the intervention is paramount for overall treatment success. Therefore, we opted for this outcome measure.
R1: Similarly, implant survival as a primary endpoint is inadequate for this design since not all groups have implant placement. Therefore, secondary measures, i.e., actual bone gain, should be the main objectives.
CA: We would like to thank the reviewer for pointing out this issue. Implant survival was moved to the secondary outcome measures and bone gain was moved to the primary outcome measures.
R1: On the other hand, the different patients' characteristics in each group are not identified in tables, nor are the data identified in tables, which is very interesting to observe. Similarly, significant differences between the different groups are not identified. This aspect is essential to contribute.
CA: We have provided raw data of the measurements in the supplementary materials (Supplementary Table 3).
We have added a table (Supplementary Table 2) to present the patient characteristics.
The following was added to the Results section:
“Patient characteristics are presented in Supplementary Table 2.”
Similarly, aspects related to ethics or aspects related to the statistical part are not indicated in the article's text.
CA: We have added the methods used for descriptive statistics to subsection 2.4.
We have added the following to the Institutional Review Board Statement: The observational retrospective study design presented in this manuscript did not require the approval of an ethics committee, as per Italian legislation on clinical investigations at the time of the study. Nevertheless, the investigation followed the rules of the Declaration of Helsinki of 1975, revised in 2013, and performed according to the principles of the ICH Good Clinical Practice.
R1: Finally, the discussion could be greatly expanded, given the large amount of literature on guided bear regeneration. It would be interesting to discuss how and in what way each of the elements that make up the FIRST technique contributes to regeneration, that is, to sustained results.
CA: The following was added to the Discussion section, to elaborate on how and in what way each of the elements that make up the FIRST technique contributes to regeneration
“GBR is characterized by using membranes to inhibit the ingrowth of connective tissue end epithelium in the space designated for bone regeneration [12-14]. Collagen membranes in combination with particulated bone graft materials have a limited space maintenance capacity, as pressure on the tissues may compress the augmented area. Therefore, such a combination of biomaterials is primarily applied for self-containing defects. Titanium-reinforced polytetrafluoroethylene (PTFE) membranes and titanium meshes combined with collagen membranes have been suggested to increase the barrier's capacity to maintain space, so that more extensive vertical and horizontal augmentations may be performed [10]. However, if opted for, titanium meshes and non-resorbable membranes have to be removed with a further surgical intervention following bony regeneration of the augmented site [10,14]. During F.I.R.S.T., a rigid bone barrier with sufficient space maintaining capacity is applied as a membrane [10-25]. The barrier and the particulated bone graft material are fixed to the bone using a fibrin sealant, guaranteeing the immobility of the site during healing [37-40]. No further surgical intervention is required as all biomaterials applied as part of F.I.R.S.T. are resorbed or integrated during healing [37-40]. According to the results of this study, F.I.R.S.T. can be used successfully for GBR of non-self-containing defects, such as ARP and immediate implant placement in case the bone walls are compromised, vertical, and extensive horizontal augmentation.”
CA: Once again, we thank the reviewer for suggesting corrections to our manuscript. Adding a subsection for the descriptive statistical analysis, revising the abstract, identifying further avenues of research, and adding a Table for Patient characteristics improve the quality of our manuscript. We hope that the reviewer finds the changes made sufficient, and recommends our manuscript for publication in this esteemed Journal.
Reviewer 2 Report
Comments and Suggestions for Authors
ABSTRACT
It is well summarized and reflects the study clearly and objectively.
INTRODUCTION
The authors have provided a good review of the state of the art in terms of techniques and materials. The introduction is very well structured and clear and objective.
MATERIALS AND METHODS
Study design - It is well done. The techniques are well presented and well documented with good quality images - however there are photographs that should be reviewed as they are presented with transversal lines.
Authors must clarify which statistical analysis was chosen and the reason for this choice.
RESULTS
The results reflect the methodology used for the study carried out. They are enlightening and scientifically acceptable.
DISCUSSION
The discussion covers several studies carried out, related to the work carried out by the authors. These are updated and published in scientifically valuable journals. The authors compare their results with those of these authors.
The limitations of the study were addressed.
CONCLUSIONS
The conclusions are objective and clear. They are short and related to the results obtained.
REFERENCES
The references are appropriate for the study. The authors have carried out an exhaustive and well-updated review.
Author Response
Corresponding author (CA): We would like to thank the reviewer for their time and effort in reviewing our manuscript and pointing out the issues that needed corrections.
Reiewer 2 (R2) ABSTRACT
It is well summarized and reflects the study clearly and objectively.
INTRODUCTION
The authors have provided a good review of the state of the art in terms of techniques and materials. The introduction is very well structured and clear and objective.
MATERIALS AND METHODS
Study design - It is well done. The techniques are well presented and well documented with good quality images –
CA: Thank you very much!
R2: however there are photographs that should be reviewed as they are presented with transversal lines.
CA: We have revised the photographs to follow Journal Guidelines. However, we have found that in certain internet browsers these transversal lines appear.
R2: Authors must clarify which statistical analysis was chosen and the reason for this choice.
CA: We have added a subsection to the Materials and Methods section to explain descriptive statistics performed for this study:
“2.4. Statistical analysis
The quantitative data were expressed as standard descriptive statistics. The Shapiro-Wilk-test was performed to assess whether data were normally distributed. For outcome measures that follow a normal distribution, the means and standard deviations were reported. In case of non-normal distribution of the data, the median and range were displayed. Descriptive statistical analyses were conducted using SPSS 25.0 software (IBM Corporation, New York, United States). Descriptive statistics were corrected throughout the manuscript.”
We have revised the results accordingly.
R2: RESULTS
The results reflect the methodology used for the study carried out. They are enlightening and scientifically acceptable.
DISCUSSION
The discussion covers several studies carried out, related to the work carried out by the authors. These are updated and published in scientifically valuable journals. The authors compare their results with those of these authors.
The limitations of the study were addressed.
CONCLUSIONS
The conclusions are objective and clear. They are short and related to the results obtained.
REFERENCES
The references are appropriate for the study. The authors have carried out an exhaustive and well-updated review.
CA: Thank you very much!
CA: Once again, we thank the reviewer for suggesting corrections to our manuscript. Adding a subsection for the descriptive statistical analysis improves the quality of our manuscript. We hope that the reviewer finds the changes made sufficient, and recommends our manuscript for publication in this esteemed Journal.
Reviewer 3 Report
Comments and Suggestions for Authors
Dear authors,
I have several suggestions and comments:
1. please include the Ethical Committee and number of approval for this study
2. the surgical procedure as explained in section Materials and methods is not the vertical augmentation, it is just the horizontal bone augmentation, while vertical relation can be achieve only using rigid metal membranes, like titanium mesh or titanium reinforced membranes or even using Urban's technique but only with the specific way of fixation resorbable membrane, not just covered with it. So, please change the whole manuscript and delete vertical, and leave just alveolar ridge preservation and horizontal augmentation indications for you study.
3. immediate implant placement indication- you wrote that the procedure was done by raising full-thickness mucoperiosteal flap, and on the clinical photos you have shown completely different situation
4. which is the difference in indication setting in between radiographs on 3b and 4b, and you did it using two different methods and set the patients in two different study groups??
5. in the modern dental implantology it is not recommended to measure the difference between preoperative and postoperative bone volume manually, and should be done using digitally based volumetric measurements.
Author Response
Corresponding author (CA): We would like to thank the reviewer for their time and effort in reviewing our manuscript and pointing out the issues that needed corrections.
Reviewer 3(R3): I have several suggestions and comments:
- please include the Ethical Committee and number of approval for this study
CA: We have added the following to the Institutional Review Board Statement: The observational retrospective study design presented in this manuscript did not require the approval of an ethics committee, as per Italian legislation on clinical investigations at the time of the study. Nevertheless, the investigation followed the rules of the Declaration of Helsinki of 1975, revised in 2013, and performed according to the principles of the ICH Good Clinical Practice.
R3: 2. the surgical procedure as explained in section Materials and methods is not the vertical augmentation, it is just the horizontal bone augmentation, while vertical relation can be achieve only using rigid metal membranes, like titanium mesh or titanium reinforced membranes or even using Urban's technique but only with the specific way of fixation resorbable membrane, not just covered with it. So, please change the whole manuscript and delete vertical, and leave just alveolar ridge preservation and horizontal augmentation indications for you study.
CA: As we have pointed out throughout the manuscript, the fibrin sealant applied to the particulated xenograft provides it with a putty consistency. The membrane used in this study is a rigid lamina produced from porcine cortical bone. Therefore, it not only acts as a cell-occlusive barrier, but also maintains space in the long-term, protecting the particulate xenograft. It is fixed to the bone using the fibrin sealant. Therefore, FIRST enables the clinician to augment the alveolar ridge vertically. This retrospective study evaluates the vertical bone gain that may be achieved by this surgical methodology. Therefore, we have not deleted the vertical GBR group from the manuscript.
We agree that further multicenter randomized clinical trials are required to compare the effectiveness of FIRST for vertical GBR to more established methods that the reviewer mentioned. Therefore, we have revised the Discussion section:
A potential area for further research is conducting multi-center randomized clinical trials that compare FIRST to traditional GBR to assess its effectiveness.
R3:3. immediate implant placement indication- you wrote that the procedure was done by raising full-thickness mucoperiosteal flap, and on the clinical photos you have shown completely different situation
CA: We have revised the text: The surgical site was accessed using the tunnel technique.
R3: 4. which is the difference in indication setting in between radiographs on 3b and 4b, and you did it using two different methods and set the patients in two different study groups??
CA: In subsection 2.2.2. we have explained how we have decided on ARP or immediate implant placement:
“In cases where the available bone beyond the apex of the tooth was sufficient and primary stability could be achieved, atraumatic extraction was performed, and immediate implant placement was carried out. “
In the case of Figure 4, we have deemed that there is no sufficient bone apically to the tooth to achieve primary stability. In this specific case even if there was sufficient bone above the apex, the implant could not be placed in the prosthetically correct position, therefore, we have decided to perform ARP and delayed implant placement.
We have revised the text to make the decision-making process clear:
“In cases where the available bone beyond the apex of the tooth was sufficient, primary stability could be achieved, and the implant could be placed in the prosthetically correct position, atraumatic extraction was performed, and immediate implant placement was carried out. In the absence of either of these criteria, ARP and delayed implant placement were performed.”
R3: 5. in the modern dental implantology it is not recommended to measure the difference between preoperative and postoperative bone volume manually, and should be done using digitally based volumetric measurements.
CA: Automatic volumetric measurements are indeed quick and reliable in measuring the volume changes of bony structures. However, for the practicing implantologist, linear measurement may be more informative, as it presents the horizontal and vertical bone gain that may be achieved using FIRST specifically at the sites of implant placement. Volumetric measurements show the bone gain overall, regarding the entirety of the augmented area, and are less informative to the clinician. Therefore, in this study, we have opted against measuring the bone volume changes.
Nevertheless, we have added the following to the limitation section:
“Another limitation of this study is that linear measurements of HBG and VBG were performed manually. Volumetric bone changes could have been an informative outcome measure to assess. Automatic assessment of bone gain may have decreased human error.”
CA: Once again, we thank the reviewer for suggesting corrections to our manuscript. Elaborating on the limitations of our research and revising the decision-making process between immediate implant placement and ARP greatly improves the quality of our manuscript. We hope that the reviewer finds the changes made sufficient, and recommends our manuscript for publication in this esteemed Journal.
Round 2
Reviewer 1 Report
Comments and Suggestions for Authors
no more comments.
Reviewer 3 Report
Comments and Suggestions for Authors
Dear authors,
you have very nicely packaged your answers to the comments that you did not take into account, but unfortunately, the flaws in your work are still present. I insist that they be corrected in case of a decision to publish the work. Warm greetings